# The Impact of Vanilla and Lemon Aromas on Sensory Perception in Plant-Based Yogurts Measured with Static and Dynamic Methods

**DOI:** 10.3390/foods11142030

**Published:** 2022-07-08

**Authors:** Maija Greis, Roosa Kukkonen, Anna-Maija Lampi, Laila Seppä, Riitta Partanen, Mari Sandell

**Affiliations:** 1Department of Food and Nutrition, University of Helsinki, P.O. Box 66, 00014 Helsinki, Finland; roosa.kukkonen@helsinki.fi (R.K.); anna-maija.lampi@helsinki.fi (A.-M.L.); laila.seppa@helsinki.fi (L.S.); mari.sandell@helsinki.fi (M.S.); 2Valio Ltd., P.O. Box 10, FI-00039 Helsinki, Finland; riitta.partanen@valio.fi

**Keywords:** aroma, texture, mouthfeel, cross-modal interactions, plant-based yogurt, temporal dominance of sensations

## Abstract

The application of cross-modal interaction is a potential strategy to tackle the challenges related to poor sensory properties, such as thin mouthfeel, in plant-based yogurts. Thus, we aim to study the influence of aroma compounds possibly congruent with sweetness on the perceived sensory profile. Descriptive analysis and temporal dominance of sensations (*n* = 10 × 4) with a trained panel were conducted with and without a nose clip. One unflavored sample and samples flavored with either lemon or vanilla aromas were included (vanilla; 0.05%; 0.1%; lemon: 0.025%; 0.05%). Odor intensity, thick, sticky, and melting sensation, sweetness, and grain-like flavor were evaluated on an unstructured 10-cm line scale with anchors and reference samples. The results demonstrate how vanilla and lemon aromas suppressed grain-like flavor and enhanced odor intensity and sweetness. The following order was detected among samples in perceived sweetness intensity: unflavored < lemon < vanilla. The two sessions with and without nose clip differed statistically in sweetness, highlighting that the aromas impacted the perceived sweetness but not the mouthfeel in vanilla samples. The study suggests that congruent aromas could modify the perceived sweetness in plant-based yogurts; however, aroma or perceived sweetness does not impact the mouthfeel in plant-based yogurts. While the odor–taste interaction in such products is evident, the study highlights that aroma compounds alone do not modify mouthfeel.

## 1. Introduction

One of the main factors holding back the more widespread adoption of plant-based products is their sensory properties [1,2,3]. In plant-based yogurts, the significant challenges are texture-related; thickness and creaminess have been found to be the main drivers of liking, and wateriness and thinness the main drivers of disliking [4,5,6,7,8]. Given the importance of texture properties in plant-based yogurt alternatives among consumers, the food industry needs new methods to tackle the challenges of mouthfeel perception. Cross-modal interaction (aroma–taste–texture) is a potential strategy, and thus, should be evaluated with such products.

Cross-modal interactions have been widely studied with aroma and taste integration to understand flavor. It has been previously discovered how the taste and odor intensities are stronger when the pair is harmonious or are typically encountered together, and thus, associative learning has been shown to play a role in the interaction [9,10]. For example, vanilla aroma has been continuously found to increase sweetness in model and natural foods [9,11,12,13,14,15,16]. Namely, cognitive cross-modal interaction between vanilla and sweetness among Chinese and Danish consumers was presented in [16]. In contrast, lemon aromas have been found to enhance sweetness or sourness depending on the food matrix and aroma compounds [12,17].

Texture-related cross-modality is less understood, and it primarily focuses on how texture affects either odor, taste, or flavor perception. Increasing viscosity, e.g., by adding hydrocolloids, generally decreases aroma and taste perception, resulting from lower volatile mobility at the food–air interphase [18]. The same has been demonstrated in many food matrices [19,20,21,22,23,24,25,26,27]. In contrast, studies on the impact of aromas on texture perception are contradictory and seem to vary depending on the aroma compound and food matrix. Moreover, most results cover mainly thickness but do not focus on other mouthfeel properties. Some interactions have been proved with varying aroma compounds and foods [28,29,30,31,32,33]. For instance, it was demonstrated by Saint-Eve et al. how fruity aroma decreased thickness perception in low-fat stirred yogurts [32]. In addition to the aroma–texture interactions, many of the studied interactions consider the taste–texture relationship [14,23,34,35]. Moreover, the sweetness–texture interaction in model dairy desserts was studied by Lethuaut et al. They observed interaction with sucrose and thickness perception, but no common rule was applied [23]. 

Most studies have been conducted with static methods, and fewer have focused on dynamic sensory perception [34,35,36,37]. More attention should be paid to the dynamics of sensory perception, as sensory interactions arise due to the simultaneous perception of different stimuli [38]. Mainly when aroma–texture-related sensory interactions are studied, Temporal Dominance of Sensation (TDS) could provide essential information on the order of perceived attributes, as aroma compounds are released due to oral processing. Moreover, Saint-Eve et al. concluded that time-dependent methods could better help understand the relationship between cross-modal interactions. They studied the texture–aroma interaction in model candies, using TDS associated with descriptive analysis [34]. A review of aroma perception in dairy products showed that temporal methods have the advantage of providing accurate descriptions of aroma perception [39]. In another work, TDS showed its potential to underline aroma–taste interaction during coffee drinking by pairing TDS with in vivo nose-space measurements [40]. In a recent study, Velázquez et al. used TDS to study the sugar reduction in vanilla milk deserts in children [35]. 

Two presumptions in the related literature are included in the study: (1) associative learning between odor and taste is dependent on the specific aromas and (2) sweetness has been shown to enhance perceived thickness in semisolids. Therefore, this study aimed to determine the influence of aroma compounds possibly congruent with sweetness on perceived sensory profile, particularly on the mouthfeel properties. We studied this question with both static and dynamic methods.

## 2. Materials and Methods

### 2.1. Samples

The unflavored fermented oat-based yogurt base was obtained from Valio (Valio Ltd., Helsinki, Finland) for the experiment. The fermented and pasteurized oat base included water, oat flour, sugar, pea protein, canola oil, calcium, NaCl, B2, B12, D2 vitamins, and iodine. Two commercial mixtures of natural aroma compounds described as vanilla and lemon were obtained from Valio (Valio Ltd., Helsinki, Finland). Different levels of concentrations were tested in preliminary experiments (*n* = 3). The original applicable range for concentrations was obtained from Valio. Finally, low and high aroma concentrations were included in the evaluation to determine if the aroma’s intensity would impact the samples’ sensory properties (Table 1). In order to be systematic in both aromas, the high concentration was double the low concentration. The samples were prepared in 500 g batches. Aroma mixtures were added to the flavored samples by Thermo Fisher pipette (Thermo Fisher Scientific, Waltham, MA, USA), and flavored and unflavored samples were then mixed at room temperature with Kenwood Cooking Chef mixer Model KM800, Finland, (Kenwood Ltd., Havant, UK) for 1 min in level 1 out of 7 (the maximum *p* = 1000 W for level 7). The viscosity was later measured to make sure the added aromas did not affect the structure of the samples (Section 2.3).

### 2.2. Sensory Analysis

#### 2.2.1. Panelists

The sensory panel was recruited from the Department of Food and Nutrition. The panel consisted of 10 members (nine women, one man) aged 24–29 years (mean 25 years). Before entering the panel, the ability to identify basic tastes modalities and describe different aromas was tested [41,42]. The study followed the ethical requirements of the sensory laboratory approved by the University of Helsinki Ethical review board in humanities and social and behavioral sciences, and the panelists signed informed consent before the evaluation started. 

#### 2.2.2. Procedure

The development of the vocabulary and evaluation techniques was carried out during training sessions (a total of 5.35 h). During the training sessions, the panelists (*n* = 10) individually compared the unflavored samples and samples with different vanilla or lemon aromas levels. Subsequently, the panelists suggested and approved the descriptors and definitions during in-group discussions. References, if possible, were proposed and agreed upon. The panelists also practiced the evaluation with a nose clip during the training sessions. Nose clips were included in the study to detect the possible impact of aromas on mouthfeel among the flavored and unflavored samples. 

The evaluation procedure consisted of generic descriptive analysis (GDA) and temporal dominance of sensation (TDS) questions. Both methods were conducted with Red Jade sensory software (RedJade Sensory Solutions LLC, Martinez, CA, USA). The evaluation was repeated four times (*n* = 10 × 4). In addition, each evaluation session was conducted twice, once with a nose clip and once without a nose clip. In order to avoid an order effect in the experiment, the nose clip condition was balanced between panelists [14]. Half of the panel began with the nose clip condition, followed by the without nose clip, while the other half worked in the reverse order. Furthermore, the sample order was also balanced between panelists and sessions. The samples were coded with three-digit codes, and their order was randomized for each assessor and session. The samples had different codes for the evaluations with and without nose clips. A total of 30 g of each sample was used for the evaluations, and they were evaluated at 12 °C. Evaluations were conducted in individual booths under green light. Panelists were instructed to cleanse their palates with tap water and unflavored corn snacks between the samples.

The intensity of odor, mouthfeel, taste, and flavor characteristics were evaluated on an unstructured, 10-cm line scale with anchors (Table 2). In total, six attributes (aroma intensity, thickness, stickiness, melting sensation, sweetness, and grain-like flavor) were included in the analysis. The evaluations with and without nose clips were identical, except that the odor and flavor-related questions were not included in the nose clip evaluation. 

In order to familiarize the panelists with the TDS method, a warm-up session with oat-based yogurt was organized in the last training session. TDS questions were asked after the GDA. In total, seven attributes, including thickness, stickiness, melting sensation, sweetness, grain-like flavor, vanilla flavor, and lemon flavor, were included in the TDS. Panelists were instructed to evaluate the samples’ sensory properties and select the term that caught their attention at each moment of the evaluation [43]. They were instructed to consume the sample for at least 5 s and no more than 30 s. The software showed the clock during the evaluation.

### 2.3. Instrumental Measurements

Color and rheological measurements were conducted to detect any possible color or physical textural differences between the samples. The volatile compounds of the unflavored and flavored yogurt samples were detected to understand the volatile profile between the flavored and unflavored samples.

The color was analyzed from each sample to demonstrate that the aroma addition did not have an impact on the color of the samples. The color was analyzed from the surface of the sample by a Minolta colorimeter (Minolta CR-400 H3063, Konica Minolta, Tokyo Japan). The color values L*, a*, and b* are reported from three replicates (20 g). 

Apparent viscosities were calculated through flow curves (FC) to demonstrate the impact of aroma addition to the texture properties. Cone-plate configuration (cone-diameter 35 mm, angle 2°, gap 0.100 mm) was used for the measurements. Flow curves (FC) were obtained from the stepped shear stress ramp between 0.01 s^−1^ and 300 s^−1^. The consistency index K and shear thinning index *n* were calculated using the Power law equation based on the flow curves. Apparent viscosities (ηapp) at shear rates 10 (s−1) were calculated from the upward flow curve (Pa·s) from Ostwald–de Waele ƞapp = k(dγ/dt)*n*−1. The measurements were carried out with HAAKE MARS 40 Rheometer and monitored by a RheoWin software package, version 2.93 (Thermo Fisher Scientific, Waltham, MA, USA).

Analysis of the volatile compounds of yogurts was performed using headspace solid-phase microextraction coupled to gas chromatography with mass spectrometric detection (HS-SPME-GC-MS), as presented previously by Damerau et al. [44]. In short, 2 g of the yogurt samples were analyzed in triplicate. The sample vials were stabilized at 60 °C for 20 min (with agitation of 250 rpm), and volatile compounds were extracted at 60 °C for 30 min (with agitation of 250 rpm) and collected to a divinylbenzene/Carboxen/polydimethylsiloxane fiber (50/30 µm film thickness; Supelco, Bellefonte, PA, USA). The volatile compounds were analyzed using an SPB-624 (30 m × 0.25 mm i.d., 1.4 µm film thickness; Supelco, Bellefonte, PA) column by GC-MS. All other GC-MS parameters were similar to those earlier [44], except the scan range was modified to *m*/*z* 40–300.

### 2.4. Data Analysis

Data from the descriptive analysis were subjected to a normality test, which showed normal distribution. A one-way ANOVA was applied to identify significant differences among samples. Tukey’s HSD (α = 0.05) multiple comparison tests tested possible differences between samples. A three-way ANOVA was applied to study the significant levels of the main effects (sample, panelist, and session) and interaction effects (sample by session, panelist by sample, and session by panelist). The differences between sessions with and without nose clips were tested by paired t-test among all the samples using a Bonferroni-corrected *p*-value (α = 0.01). Correlations between sensory characteristics were determined using Pearson’s Correlation among all the samples (*n* = 40 × 5). 

For the TDS data, the order of the dominant attributes was obtained by plotting the dominance rates of each of the sensations at different time points of the eating period, as described earlier in Greis et al. 2020. The data from each subject were standardized across the whole eating time. Only the dominant rates higher than the significance level were used for the figures. The significance level was calculated as proposed by [43].

Instrumental measurements were taken in triplicate, and the results are presented as mean value with standard deviation. All the analyses were conducted using the SPSS Statistics 24.0 program (SPSS Inc., Chicago, IL, USA). Volatile compounds were analyzed in the headspace of three different yogurts (U-0, V-2, and L-2). The mean of the relative proportions and standard deviation were calculated for each sample and volatile compound.

## 3. Results

### 3.1. Sensory Analysis

#### 3.1.1. Generic Descriptive Analysis

When the nose clip was not used, clear differences between the samples were found in aroma intensity (F(4.35) = 31.6, *p* < 0.001), grain-like flavor (F(4.35) = 38.9, *p* < 0.001) and in sweetness (F(4.35) = 2.5, *p* = 0.041) (Figure 1). Samples with lemon aromas obtained the highest scores in aroma intensity and lowest scores in grain-like flavor of all the samples. Moreover, sample V-2 was evaluated as the sweetest among the samples. No differences were found between the samples in any of the attributes when the nose clip was used. Furthermore, a significant effect of the interaction of panelist × sample and panelist × replicate was found in some of the attributes (Appendix A). 

The evaluation condition, nose clip, affected the results only in sweetness (Table 3). In sweetness, differences were found in sample V-2: t(39) = 2.815, *p* < 0.008 with the Bonferroni corrected *p*-value (0.01). The sample was evaluated as sweeter when the nose clip was not used. Correlations between sensory properties among all the samples are presented in Table 4. Aroma intensity and grain-like flavor correlated negatively, whereas melting sensation correlated positively with both thickness and stickiness. Sweetness correlated positively with all three mouthfeel attributes and grain-like flavor.

#### 3.1.2. Temporal Sensory Profile

The TDS task captured the temporal evolution of the attributes for all the samples. The temporal sensory profile indicates the order of the significantly dominant attributes (Figure 2). The results suggest that when the nose clip was on, the mouthfeel properties were more highlighted, as thickness, stickiness, and melting sensations got the highest significant citation proportions, in respective order. The flavor properties dominated without the nose clip, apart from thickness, which had a lower citation proportion in vanilla samples than lemon samples. Moreover, the sweetness was dominant only in vanilla samples. Furthermore, the results demonstrate that vanilla and lemon aromas have also been detected with a nose clip, particularly at the end of the mastication.

### 3.2. Instrumental Measurements

The apparent viscosity (10 s^−1^) was identical among samples ranging from 3.37 to 3.45 (Table 5). Moreover, the color was measured to be similar between the samples. All the samples showed shear thinning behavior (*n* < 1) as the apparent viscosity decreased by increasing the shear rate in all the samples.

The volatile profile of the three different samples was detected to be very different; the U-0 sample had the lowest numbers, while the V-2 had the highest number of detected compounds (Table 6). The volatile compounds found in the U-0 sample were the same as in the other two flavored samples apart from the acetic acid, which was found only in the U-0 sample, and nonanoic acid, which was found only in the U-0 and V-2 samples. In total, 6 out of 16 aroma compounds were found only in the V-2, whereas 31 out of 35 compounds were found only in the L-2.

## 4. Discussion

The impact of added aromas on the sensory profile of the yogurt samples was evident. The GDA demonstrates how vanilla and lemon aromas enhanced odor intensity and perceived sweetness and suppressed grain-like flavor. The results are similar to the TDS results; the added flavors, particularly lemon flavor, dominated the flavor profile over grain-like flavor and sweetness. The aroma intensity was highest, and the grain-like flavor was lowest in the lemon sample. The volatile profile of the samples explains this; the volatile profile was simpler in unflavored and vanilla samples than in lemon samples, as fewer aroma compounds were found in them. Furthermore, the aroma compounds with the highest proportion were vanillin in the vanilla sample and Dl-limonene in the lemon sample with vapor pressures of 1.18 × 10^−4^ mmHg and 1.55 mmHg at 25 °C, respectively [45,46]. The higher vapor pressure of Dl-limonene makes it more volatile in the mouth compared to vanillin.

According to the GDA and TDS results, the intensity of grain-like flavor was highest in the unflavored sample and lowest in both lemon samples, indicating that the lemon aromas suppressed the grain-like flavor more effectively than vanilla aromas. All three samples shared three volatile compounds: 2,3-butanedione and 2,3-pentanedione, contributing to creamy and buttery flavors, and 3,5-octadiene-2-one, contributing to a fruity, green, and grassy flavor [47,48]. In addition, 2-pentylfuran has been connected with grass and raw flavors in oat flour [49], and 1-hexanol has been associated with cut grass flavor in oat groats [47]. The unflavored sample had a higher volatile proportion associated with grass-like flavor than vanilla and lemon samples, supporting the more intense grain-like flavor in these samples. For instance, 2-pentylfuran has a very low threshold value and is, thus, an essential contributor to flavor [50,51]. The volatile profiles suggest that the added aromas did not only mask the grain-like flavor, but the grassy-like aroma compounds were more present in unflavored samples. This could be due to 1-hexanol and 2-pentylfuran reacting with other flavor compounds in the matrix. Covering all the compounds that contribute to grain-like flavor is challenging because their flavor thresholds vary dramatically. Even minor differences may have a considerable impact [51]. 

The following order between the samples was detected in perceived sweetness intensity: unflavored < lemon < vanilla. Moreover, the sessions with nose clip and without nose clip differed statistically, only in perceived sweetness, demonstrating that the aromas impacted the taste. Previously, it has been demonstrated that lemon aromas can enhance sweetness or sourness, depending on the aroma compounds and food matrix [12,17]. In this study, it was evident that the lemon aroma compounds enhanced perceived sweetness compared to the unflavored samples. TDS results indicate, however, that the lemon flavor dominated sweetness. 

The results are consistent with previous literature on vanilla aroma, which has been continuously found to increase sweetness due to associative learning [15,16,24]. While GDA results indicate that vanilla aromas enhanced sweetness, TDS results also demonstrate the same: sweetness and vanilla flavor were evaluated at the beginning of the evaluation, highlighting that the interacting properties are perceived simultaneously [38]. Furthermore, it has been captured by Charles et al. that fruity aroma enhances sweetness at the beginning of the evaluation in apples [37]. They hypothesize that volatile compounds are released quickly from the apple tissue, eliciting the sweetness. This emphasizes that the interaction’s timing depends on when the volatile compounds are released, which, in contrast, depends on the aroma compound, its vapor pressure, and the food matrix composition [39,52,53]. Our results also highlight that the first period of mastication is essential for sensory interaction. Thus, the time dimension should be considered when evaluating the aroma–taste and aroma–texture interactions. 

Only insignificant yet logical differences could be seen in mouthfeel properties: vanilla samples scored higher in thickness and stickiness. Furthermore, they melted slower than lemon and unflavored samples. Interestingly, according to the TDS results, sweetness and thickness were selected around the same period, indicating that the time mechanism would support the potential interaction. This indicates that sweetness and thickness could interact in another food matrix. Significant and positive Pearson correlations also support the possible connection between sweetness and mouthfeel properties. 

Our results support the previous findings that the impact of aroma–taste–texture interaction depends on the aroma compound [39]. The results are consistent with some of the previous literature on semisolids. For example, a study by Tournier et al. explored aroma or taste interactions with texture in custard desserts, varying in viscosities, sucrose level, and aroma nature. They did not find aroma or taste interaction with texture [14]. In contrast, a study by de Wijk et al. concluded that vanilla aromas resulted in thickness, creaminess, and fattiness in vanilla custards [31]. Moreover, Kora et al. demonstrated how fruity aroma decreased thickness in low-fat stirred yogurts [30]. 

A previous study by Saint-Eve et al. suggests how yogurts flavored with a mixture of aromas were perceived as thinner and smoother than those flavored with only one aroma compound [32]. A similar trend is seen in our TDS results, as the vanilla sample had a smaller number of volatiles present. The vanilla samples were not perceived as thick as the lemon samples (without the nose clip). The results are in line with the similar yet insignificant GDA results. Thus, further research is needed to prove the impact of the number and complexity of aroma compounds on the mouthfeel.

### Reliability of the Methods

The apparent viscosity demonstrated how the added aroma did not affect the samples’ physical texture; thus, it was the same for the flavors to release. While the impact of the aroma–taste–texture interaction was presumably small, a generic descriptive analysis with a trained panel proved to be a better choice than a consumer test. As we found no impact on the mouthfeel with a trained panel, presumably, the differences would not be found with a consumer study either. The nose clip condition brought substantial evidence to the results between the unflavored and flavored samples. TDS results demonstrated, however, that some aromas could be perceived even when the nose clip was on. This highlights that the nose clip did not block the nasal cavity entirely. Furthermore, we cannot compare the methods, as the panel completed both methods in the same order. 

## 5. Conclusions

The application of cross-modal interaction is a potential strategy to tackle the challenges related to poor sensory properties, such as thin mouthfeel, in plant-based yogurts. Cross-modal interactions, including texture, have been under less investigation than interactions including odor or taste, partly due to the complexity of texture properties. We aimed to study the influence of aroma compounds possibly congruent with sweetness on perceived sensory profile. Thus, vanilla and lemon aromas were chosen for the study. Static and dynamic methods were used to determine the time dimension of possible interaction. The results demonstrate how vanilla and lemon aromas suppressed grain-like flavor and enhanced odor intensity and perceived sweetness in plant-based yogurts. The following order between the samples was detected in perceived sweetness intensity: unflavored < lemon < vanilla. The sessions with nose clip and without nose clip differed statistically, only in perceived sweetness, demonstrating that the aromas had an impact on the perceived sweetness but not on mouthfeel properties. The study highlights that aroma compounds alone in plant-based yogurts do not easily modify mouthfeel, but aromas or aroma mixtures enhance the perceived sweetness at the beginning of mastication in plant-based yogurts. Thus, when developing healthier and tastier plant-based alternatives, the time dimension should be considered in the aroma–taste and aroma–texture interactions.

## Figures and Tables

**Figure 1 foods-11-02030-f001:**
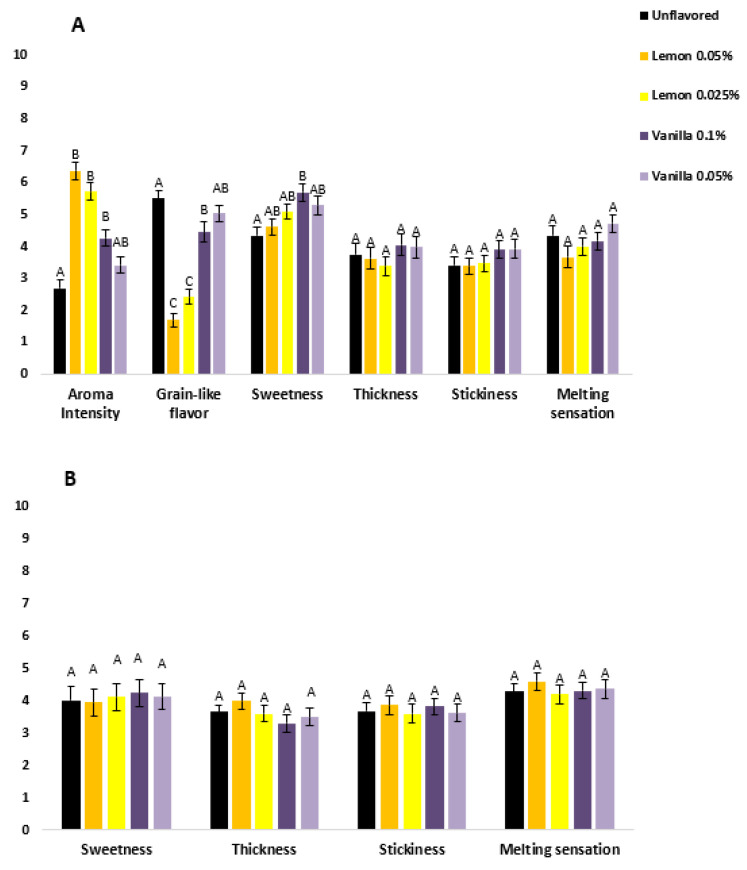
Mean ratings and the standard error of the mean (SEM) (*n* = 4 × 10) of odor, flavor, and mouthfeel properties without a nose clip (**A**) and with a nose clip (**B**) in different samples, i.e., U-0, L-2, L-1, V-2, and V-1, referring to unflavored, Lemon 0.05%, Lemon 0.025%, Vanilla 0.1%, and Vanilla 0.05%, respectively. Letters indicate a statistically significant difference between the samples: *p* < 0.05 by Tukey HSD test.

**Figure 2 foods-11-02030-f002:**
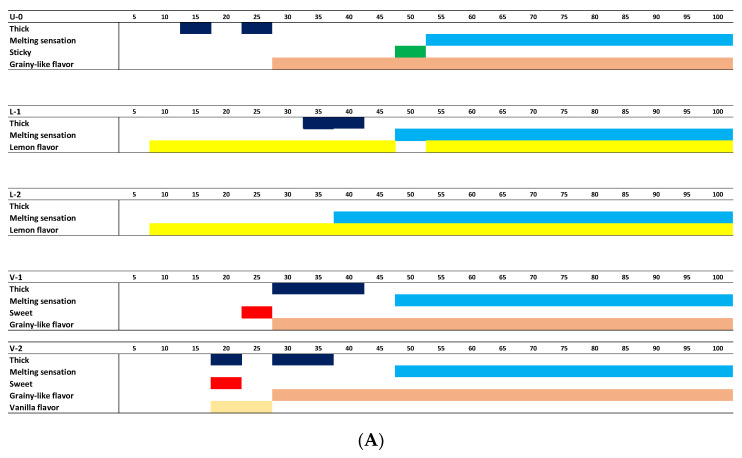
Dominant sensory properties during eating on a standardized time scale (0–100) evaluated without a nose clip (**A**) and with a nose clip (**B**) (*n* = 4 × 20).

**Table 1 foods-11-02030-t001:** Yogurt samples, added aroma or water concentration, and mixing time.

Code	Sample	Aroma or Water Concentration	Mixing Time
U-0	Unflavored	Unflavored, 0.05% water	1 min
L-1	Lemon flavored	0.025% lemon	1 min
L-2	Lemon flavored	0.05% lemon	1 min
V-1	Vanilla flavored	0.05% vanilla	1 min
V-2	Vanilla flavored	0.1% vanilla	1 min

**Table 2 foods-11-02030-t002:** The rated attributes in Descriptive analysis (GDA) and Temporal Dominance of sensations (TDS), evaluation instructions, and reference samples.

Attribute	Instructions	Reference Sample for the Attribute	GDA	TDS
**Odor**				
Odor intensity	Open the lid and sniff the sample two times.	N/A	× *	
**Mouthfeel**				
Thickness	Take a spoonful of the sample in your mouth and press it against your palate. Then, evaluate the thickness before masticating the sample.	Thin: Valio unflavored Oddlygood oatgurt base, mixed 3 min (scale value 0). Thick: Valio unflavored Oddlygood oatgurt base, unmixed (scale value 10).	×	×
Stickiness	Take a spoonful of the sample in your mouth. Evaluate stickiness after rotating the sample in your mouth twice (by rotating your tongue) and brining your tongue down from the palate.	Sticky: Fazer Aito, fermented oat yogurt, (scale value 10)	×	×
Melting sensation	Melting sensation of the sample during eating, from start to finish, on a scale from fast to slowly thinning.	Fast melting: Mö oat yogurt vanilla (scale value 0). Slowly melting: Fazer Aito, fermented oat yogurt (scale value 10)	×	×
**Flavor**				
Grain-like flavor	Take a spoonful of the sample in your mouth. Evaluate the grain-like flavor after rotating the sample two times in your mouth.	N/A	× *	×
Sweetness	Take a spoonful of the sample in your mouth. Evaluate the grain-like flavor after rotating the sample two times in your mouth.	Sweet: 1.5% Sucrose solution (scale value 10)	×	×
Lemon flavor	N/A	N/A		×
Vanilla flavor	N/A	N/A		×

* This attribute was not evaluated with nose clip. N/A = Not applicable.

**Table 3 foods-11-02030-t003:** Mean and standard deviation in thickness, stickiness, melting sensation, and sweetness among all the samples (*n* = 40 × 5). The *p*-value indicates the difference between evaluations with and without nose clip (NC) among all the samples by paired *t*-test.

Attribute	Mean with NC	Mean without NC	*p*-Value
Thickness	3.6 ± 1.6	3.8 ± 2.2	0.306
Stickiness	3.7 ± 1.7	3.6 ± 2.1	0.454
Melting sensation	4.3 ± 1.7	4.2 ± 1.7	0.224
Sweetness	4.1 ± 2.6	4.2 ± 1.9	<0.001

**Table 4 foods-11-02030-t004:** Pearson correlations between the attributes among all the samples (*n* = 40 × 5).

	Aroma Intensity	Thickness	Stickiness	Melting Sensation	Grain-Like Flavor	Sweetness
Aroma intensity	1	0.008	0.027	−0.077	−0.360 **	0.108
Thickness		1	0.600 **	0.542 **	0.087	0.310 **
Stickiness			1	0.287 **	0.058	0.355 **
Melting sensation				1	0.104	0.245 **
Grain-like flavor					1	0.202 **
Sweetness						1

** Correlation is significant at the 0.01 level (2-tailed).

**Table 5 foods-11-02030-t005:** Physical properties of yogurt samples. Apparent viscosity has been calculated from three separate measurements at a shear rate 1 (s^−1^) from the upward flow curve (Pas) with the Ostwald–de Waele equation. Color measurement means and standard deviations are based on three measurements per sample.

Sample	Color			Apparent Viscosity
	L*	a*	b*	10 s^−1^
U-0	29.4 ± 0.91	1.1 ± 0.04	9.2 ± 0.46	3.40
L-1	31.8 ± 0.91	1.0 ± 0.06	9.2 ± 0.23	3.44
L-2	28.9 ± 0.45	1.1 ± 0.02	9.7 ± 0.25	3.38
V-1	31.9 ± 1.62	1.0 ± 0.13	9.9 ± 1.19	3.45
V-2	29.4 ± 0.31	1.1 ± 0.01	10.1 ± 0.37	3.37

**Table 6 foods-11-02030-t006:** Volatile compounds analyzed in the headspace of three different plant-based yogurts (U-0, V-2 and L-2):  Mean of Relative Proportions ± Standard Deviation, % (*n* = 3). The volatiles are listed in ascending order of retention time.

Compound	Unflavored (U-0)	Vanilla (V-2)	Lemon (L-2)
ethanol		9.5 ± 1.0	
2,3-butanedione	15.7 ± 0.8	2.8 ± 0.3	0.1 ± 0.0
acetic acid	21.6 ± 0.0		
2,3-pentanedione	15.4 ± 1.5	3.0 ± 0.4	
3-hydroxy-2-butanone (acetoin)		0.9 ± 0.0	
1-hexanol	22.3 ± 7.0	3.8 ± 0.4	
2-heptanone	9.0 ± 1.7	1.6 ± 0.3	
α-thujene			0.1 ± 0.0
α-pinene			0.7 ± 0.1
sabinene			0.8 ± 0.1
β-pinene			6.1 ± 0.4
β-myrcene			1.5 ± 0.1
2-pentylfuran	5.4 ± 0.6	1.2 ± 0.6	
benzaldehyde	8.3 ± 0.7	2.4 ± 0.7	
α-terpinene			0.2 ± 0.0
dl-limonene			50.7 ± 1.2
sabinone			0.2 ± 0.0
1,8-cineole			0.5 ± 0.0
γ-terpinene			6.9 ± 0.2
3,8,p,menthdiene			
α-terpinolene			1.1 ± 0.0
1-octanol			0.1 ± 0.0
α-4-dimethylstyrene			0.2 ± 0.0
2-nonanone	11.6 ± 1.6	2.2 ± 1.0	
linalool			5.5 ± 0.4
nonanal		1.1 ± 0.0	0.2 ± 0.0
3,5-octadiene-2-one	6.2 ± 0.3	1.5 ± 0.3	0.1 ± 0.0
trans-limonene oxide			1.4 ± 0.2
2E,6Z-nonadienal			0.2 ± 0.0
citronella			0.2 ± 0.1
4-terpineol			0.2 ± 0.0
α-terpineol			0.6 ± 0.1
Carveol			0.1 ± 0.0
z-citral			7.4 ± 0.6
1-carvone			0.3 ± 0.1
2-undecanone		0.9 ± 0.1	
3,7-dimethyl 2,6-octadienal (citral)			11.0 ± 0.7
nonanoic acid	8.2 ± 0.0	1.9 ± 0.7	
phellandral			0.1 ± 0.0
2E,4E-decadienal		0.7 ± 0.0	
neryl acetate			1.4 ± 0.1
levandulyl acetate			0.9 ± 0.0
piperonal		2.4 ± 0.4	
α-bergamotene			0.2 ± 0.0
trans-caryophyliene			0.2 ± 0.0
β-bisabolene			0.4 ± 0.0
vanillin		65.1 ± 5.7	

## Data Availability

No new data were created or analyzed in this study. Data sharing is not applicable to this article.

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
