# Peer review of "The Impact of Vanilla and Lemon Aromas on Sensory Perception in Plant-Based Yogurts Measured with Static and Dynamic Methods"

_foods, 2022, doi:10.3390/foods11142030_

Round 1

Reviewer 1 Report

The manuscript represents a study that established a method to study the influence of aroma compounds possibly congruent with sweetness on perceived sensory profile. 

It is well written, based on current references.

The concerns about the plant-based yogurts content are highly appreciable due to the worldwide consumption of the latter.

The results are presented with many tables and figures.

 The paper is well structured throughout, and the conclusions are supported by the results. Overall the study is good and adds something new to the existing literature which may have a positive impact. 

I have several remarks: 

- TDS first appears in the text on line 64. Therefore, it should be written what Temporal Dominance of Sensation (TDS) means.

 - Table 6 shows Unflavored (U-2) instead of Unflavored (U-0). In the same table, the a, b, and g must be replaced by a, b and g. For example a-pinene, b-pinene, g-terpinene…

Reviewer 2 Report

Dear authors, the manuscript presents an interesting topic. I recommend the manuscript for a minor revision.

Specify vanilla and lemon aromas information. It is a natural extract? How diluted? How did you

Chose the concentration of aromas?

Complete the model of Kenwood C. Ch. mixer and specify rpm or any equivalent parameter of level 1.

Complete the software for TDS method.

Complete the color measurement information.

Figure 1 notes: …with and without nose clip (which is A, B part of figure? - complete)

Table 3: explain “NC” in notes. And correct the titles in table 3 and table 4, the tables are reversed.

Table 6: “The volatiles are listed in ascending order OF retention time. The volatile compounds are listed in ascending order of retention time“ – it is dual information, correct it.

Table 6: delete 0% 2,3-pentanedione and 0% 3,8-p-menthdiene in L-2 sample

„3,5-ocatdiene-2-one“ correct „3,5-octadiene-2-one“ in the text and table.

Conclusion: Complete the know-how. Why did you try vanilla and lemon aromas in oat yogurt?

Reviewer 3 Report

Dear authors,

Congratulations! I found your work very interesting and robust.

Simple corrections in the attached file.
